# Direct observation of large electron–phonon interaction effect on phonon heat transport

Jiawei Zhou[1], Hyun D. Shin[2], Ke Chen[1,3], Bai Song[1,4], Ryan A. Duncan[2], Qian Xu[1], Alexei A. Maznev[2], Keith A. Nelson[2] & Gang Chen[1✉]

As a foundational concept in many-body physics, electron–phonon interaction is essential to understanding and manipulating charge and energy flow in various electronic, photonic, and energy conversion devices. While much progress has been made in uncovering how phonons affect electron dynamics, it remains a challenge to directly observe the impact of electrons on phonon transport, especially at environmental temperatures. Here, we probe the effect of charge carriers on phonon heat transport at room temperature, using a modified transient thermal grating technique. By optically exciting electron-hole pairs in a crystalline silicon membrane, we single out the effect of the phonon–carrier interaction. The enhanced phonon scattering by photoexcited free carriers results in a substantial reduction in thermal conductivity on a nanosecond timescale. Our study provides direct experimental evidence of the elusive role of electron–phonon interaction in phonon heat transport, which is important for understanding heat conduction in doped semiconductors. We also highlight the possibility of using light to dynamically control thermal transport via electron–phonon coupling.

[1] Department of Mechanical Engineering, Massachusetts Institute of Technology, Cambridge, MA 02139, USA. [2] Department of Chemistry, Massachusetts Institute of Technology, Cambridge, MA 02139, USA. [3] Present address: School of Physics, Sun Yat-sen University, 510275 Guangzhou, China. [4] Present address: Department of Energy and Resources Engineering, and Beijing Innovation Center for Engineering Science and Advanced Technology, Peking University, 100871 Beijing, China. ✉email: gchen2@mit.edu

The electron–phonon interaction is one of the cornerstones of condensed matter physics. It is a major scattering mechanism that limits charge carrier mobility in bulk semiconductors[1], forms the basis of conventional super-conductivity[2], and contributes to optical absorption in indirect-gap semiconductors[3]. Given its paramount importance, electron–phonon interactions, and particularly their impact on electron transport, have been extensively studied, from Hall measurements of the collective interaction between electrons and phonons[4,5], to the development of angle-resolved photoemission spectroscopy that resolves electronic band structure and inter-actions with phonons through wavevector-dependent spectral linewidths[6,7]. However, electron–phonon interaction effects on phonon transport are less well characterized, both theoretically and experimentally. While past work has shown that phonons at specific wavevectors can undergo pronounced renormalization[8,9] or experience enhanced scattering[10–12] due to electron–phonon interactions, an open question is to what extent electron–phonon interactions can alter phonon heat conduction—the collective transport of phonons with a broad spectrum. In this work, we experimentally quantify the effects of optically excited free carriers on collective phonon dynamics in silicon, revealing the direct impact of electron–phonon interactions on heat transport.

Studies into the effects of electron–phonon interactions on phonons include the pioneering work of Peierls[8] and later work by Kohn[9] who showed that the interactions can dramatically change the effective interatomic forces, leading to structural instability in low-dimensional systems and significant softening of phonon frequencies at specific wave vectors for three-dimensional materials, phenomena which have been verified through inelastic neutron scattering studies on metals[13]. Alternatively, one can obtain phonon-specific information via scanning tunneling spectroscopy, where one can extract from the voltage dependence of the tunneling current the interaction of electrons with phonons at a specific frequency[14]. This method is limited to low temperatures as increased temperatures will significantly broaden phonon-associated spectral features. Acoustic attenuation experiments have been employed to study the impact of electron–phonon interactions on the damping of acoustic phonon modes, albeit with a limited range of phonon frequency from megahertz to gigahertz[10,11]. Carrier effects on optical phonons with terahertz frequencies can also be probed by transient Raman scattering[15], to investigate the population and relaxation of select phonon modes out of equilibrium[16].

Despite these studies demonstrating the importance of electron–phonon interactions in governing specific phonon states, their impacts on heat conduction have been largely overlooked. Early theoretical studies by Sommerfeld and Bethe on metals[17], and by Ziman on semiconductors[18], mostly concluded that the electron–phonon interactions only have significant impacts on phonon heat conduction when the intrinsic phonon–phonon interactions become weaker at low temperatures and no longer dominate the phonon scatterings[19,20]. This has been corroborated by experimental studies on metals at cryogenic temperatures, in which a strong magnetic field was used to 'freeze out' the electrons[21] and measure the change in the thermal conductivity[22]. Investigations into semiconductors have been conducted by measuring the thermal conductivity of samples at different doping concentrations to understand the effect of carriers on thermal transport[23], but a major difficulty here is to unambiguously separate phonon–electron scattering from scattering by the dopant impurities themselves. Alternatively, electrostatic gating can be used to introduce carriers. However, due to the short screening length, typically on the order of a few nanometers in the semiconductor at high carrier concentration[24], carriers are confined to a thin layer and cannot interact with

phonons sufficiently, and therefore mostly have a negligible impact on the heat transport.

Recent progress in thermoelectric materials has revived interest in studying electron–phonon interactions, because many good thermoelectric materials are heavily doped semiconductors with carrier concentrations in the range of $10^{19}-10^{21}$ cm$^{-3}$. In this regime, electron–phonon interactions may have a large impact on phonon transport. Experimental studies of thermal conductivity in thermoelectric materials by simultaneously fitting several phonon scattering mechanisms have indeed suggested the possibility of strong phonon scattering by electrons at room temperature[25,26]. First-principles calculations have shown that the lattice thermal conductivity can be significantly reduced at room temperature due to the phonon–electron scattering at high carrier densities[27,28]. Recent experiments employing photoacoustic spectroscopy lent further support by demonstrating strong damping of an acoustic phonon mode at ~250 GHz due to optically excited carriers[12]. However, direct experimental verification of the impact of electron–phonon interactions on heat transport has been lacking.

In this article, we describe a time-resolved optical measurement to quantify the effect of electron–phonon interactions on heat transport in a crystalline silicon membrane, and demonstrate a significant reduction in the thermal conductivity at room temperature with good agreement between experiment and first principles calculation. Building on the conventional transient thermal grating (TTG) technique[29,30] in which crossed excitation laser pulses generate a spatially periodic temperature modulation, whose diminishment due to thermal transport is monitored through time-dependent diffraction of probe laser light, we introduce an additional optical pulse to excite electron–hole pairs. By monitoring the decay of the thermal grating at different photoexcited carrier densities, we unambiguously quantify the impact of electron–phonon interactions on heat transport. Our approach rules out the effect of phonon–impurity interactions, because the carriers are introduced optically rather than by chemical doping. At a carrier concentration of around $10^{19}$ cm$^{-3}$, the electrons and holes take tens of nanoseconds to recombine, a sufficiently long time for us to observe their impact on heat transport which occurs on the same time scale. The wavelength of the excitation beam is chosen such that it generates carriers uniformly across the thickness of the silicon membrane, maximizing the interaction between the carriers and phonons throughout the volume in which thermal transport is measured. Our approach therefore overcomes the major difficulties that have previously prevented direct quantification of phonon–electron scattering impact on heat transport. We demonstrate that the thermal conductivity of silicon at room temperature is significantly reduced due to phonon–electron scattering at carrier concentrations above $1 \times 10^{19}$ cm$^{-3}$.

## Results

**Experimental set-up and contributions to TTG signal**. In our TTG experiment[29,30], two 515-nm pump beams with ~180 fs duration pulses are crossed at the sample with an angle $\theta$ (Fig. 1a and see the "Methods" section). Due to the interference between these two beams, a sinusoidal intensity profile is created at the sample surface. As the pump beam excites free carriers which subsequently thermalize and generate heat, a sinusoidal profile of heating (and thus temperature) is created inside the sample in the in-plane direction (Fig. 1b, green color indicates the sinusoidal heating by the pump beam), leading to a corresponding variation in the complex refractive index through its temperature dependence. Following transient grating generation, heat diffusion in the in-plane direction will smooth out the temperature variation. To study the transient thermal response, a continuous-wave, 532-nm probe beam is passed through the sample and diffracted by

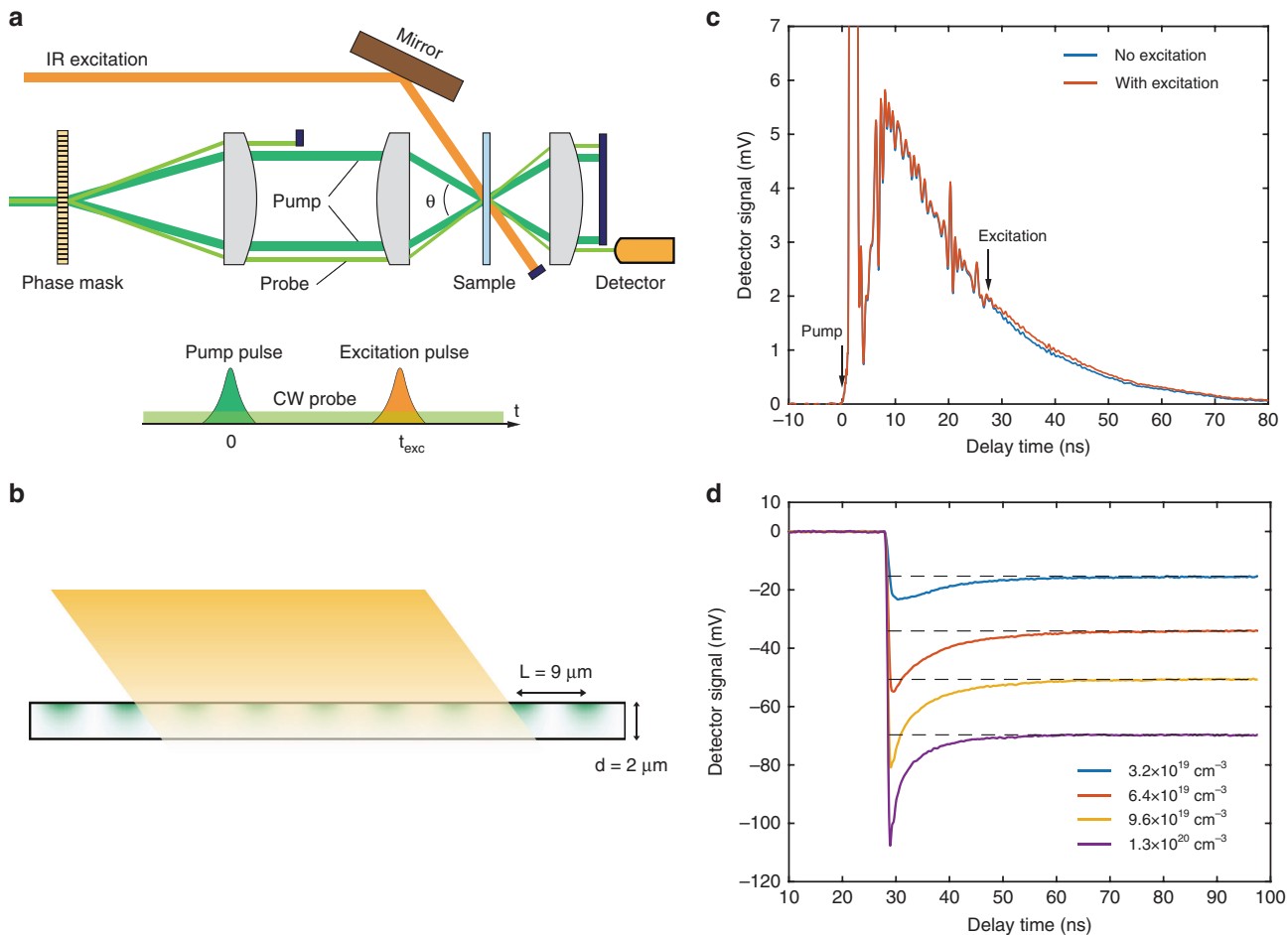

**Fig. 1 Experimental set-up and transient thermal response. a** Key components showing the pump and probe beams for transient thermal grating, and the photoexcitation beam for generating free carriers. The excitation beam is delayed relative to the pump beams by a time $t_{exc}$. **b** Illustration of the absorption of pump (green) and excitation (orange) beams in the silicon membrane. Key dimensions (grating period of 9 μm and membrane thickness of 2 μm) are indicated. **c** Representative transient thermal grating signals obtained with and without photoexcitation. The background signals (obtained with same excitation and probe configuration and without pump) are subtracted to ensure zero baseline. The excitation pulse energy was 4.0 μJ. The differences after the excitation time are clearly visible, demonstrating the impact of electron–phonon interactions on transient heat transport. **d** Transmission signal of the probe beam under photoexcitation with different excitation pulse fluences (no TTG pump pulses), which shows the recombination dynamics of photoexcited carriers. The different levels of the initial carrier density under different excitation fluences are shown. The dashed line indicates the signal level reached at longer times due to sample heating, and the difference between this and the transient absorption signal at early times represent the absorption due to photoexcited carriers.

the transient sinusoidal variation in the refractive index. The intensity of the diffracted probe beam is proportional to the square of the magnitude of the temperature variation. By measuring the diffracted probe signal as a function of time, one monitors how fast the temperature modulation decays, and thereby extracts the thermal diffusivity of the sample.

The initial spatially periodic profile of carrier density can also lead to changes in the refractive index and diffract the probe beam. As a result, besides the thermal transport, the transient grating signal also contains carrier transport information. While these two phenomena coexist, they can often be separately studied due to their different timescales. In silicon, the diffusion coefficients of electrons and holes ($10-30$ cm$^2$/s are at least one order of magnitude larger than that of thermal diffusivity ($<0.8$ cm$^2$/s) at room temperature. One can focus on the thermal transport alone by studying the signal at times sufficiently larger than the characteristic decay time of the carrier grating. A typical TTG signal obtained from a silicon membrane is shown in Fig. 1c (blue curve). The initial short spike is due to the photoexcited carrier grating which rapidly decays due to carrier

diffusion and recombination. The minimum in the TTG signal at ~4 ns is due to the fact that increased carrier density and temperature cause refractive index changes of opposite signs[31]. When the contributions from carriers and temperature balance, the refractive index variation crosses zero, resulting in a minimum in the diffraction intensity. The slow decay after 10 ns is dominated by heat transport, which is well separated from the initial fast decay due to carrier dynamics.

To study the effect of electron–phonon interactions on the heat transport, we introduce another intense optical pulse (hereafter referred to as the excitation pulse) to create free carriers in the sample (Fig. 1b), and we monitor the resulting changes in the subsequent thermal decay. The arrival time of the excitation pulse is delayed relative to the arrival of the pump pulse by 27.8 ns, which is nearly one order of magnitude longer than the characteristic decay time (~3 ns) of the carrier density grating seen from Fig. 1c. This ensures that pump-generated carrier dynamics have no contribution to the TTG signal and we are only probing the effect of excitation-generated carriers on heat transport. The excitation

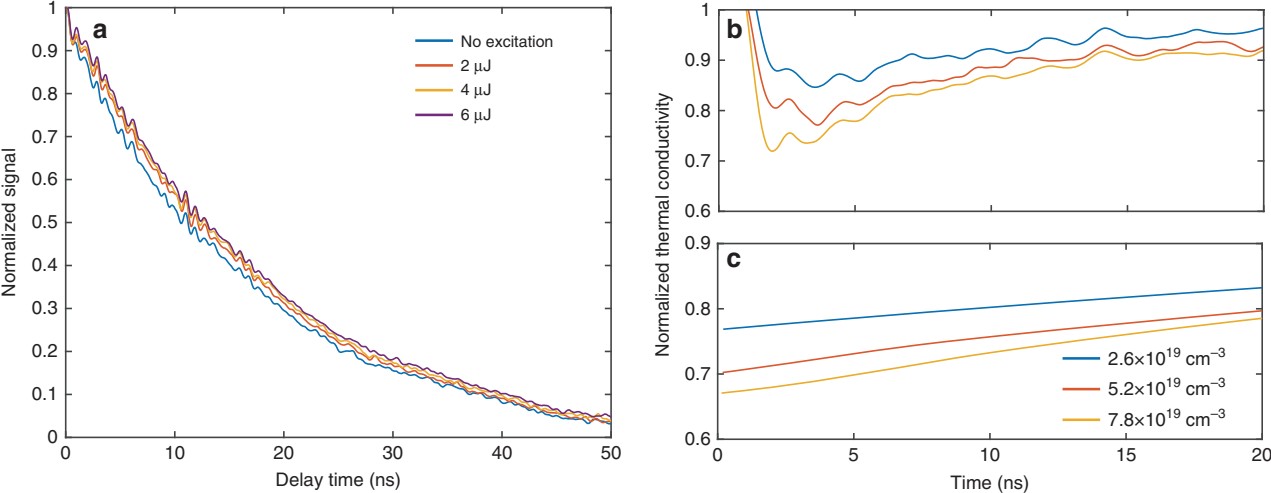

**Fig. 2 Transient heat transport with photo-excited carriers. a** Signals (after subtracting the background signals) under different excitation fluences normalized to their values at the excitation time, which is taken as the time zero. At the highest excitation pulse energy of 6.0 μJ, the initially generated density of electron–hole pairs is $7.8 \times 10^{19}$ cm$^{-3}$. The thermal decay becomes slower with higher carrier density due to the scattering of phonons by electrons and holes. **b** Normalized time-varying effective thermal conductivity extracted from experiments at three different carrier densities. **c** Normalized effective thermal conductivity obtained from model. The thermal conductivities in **b** and **c** have been normalized to the value without excitation. The general agreement between **b** and **c** indicates that the transient heat transport is governed by the carrier dynamics as phonon scattering by free carriers limits the thermal conductivity.

wavelength is chosen to be 800 nm, as the corresponding optical penetration depth in silicon is ~9.8 μm, relatively large compared to the thickness of the silicon membrane (2 μm). Furthermore, the excitation beam size (~280 μm) is considerably larger than the grating period $L$ ($L = 9$ μm in the following discussion unless otherwise noted) and the probe beam size (~100 μm). Thus the excitation beam generates carriers uniformly both laterally and across the thickness of the silicon membrane, maximizing the interaction between phonons and carriers throughout the measurement region. A uniform carrier density will not contribute to TTG signal by itself, because the diffracted signal is only sensitive to periodic variations with the lateral period $L$.

The key components of the experimental set-up are illustrated in Fig. 1a (with more details given in the "Methods" section). Briefly, the pulsed pump beam at 515 nm is split into two beams by a phase mask, and focused onto the sample surface via an imaging lens system which creates the thermal grating. The continuous-wave probe beam at 532 nm is diffracted by this thermal grating, and the diffraction intensity is monitored by a fast detector. The optical penetration depth in silicon at 515 and 532 nm is approximately 680 and 960 nm, respectively. In this transmission geometry, the signal resulting from the in-plane heat diffusion follows an exponential decay[30], with the decay time given by $\tau = \frac{\rho c_\mathrm{p}}{2q^2 k_x}$, where $\rho$ is the material density, $c_\mathrm{p}$ the heat capacity, $k_x$ the in-plane thermal conductivity, and $q = 2\pi/L$ the grating wave vector with $L$ being the grating period. The signal in Fig. 1c after 25 ns. without photoexcitation can be well fitted with an exponential curve, giving an effective thermal conductivity of ~105 W m$^{-1}$K$^{-1}$ (Fig. S1). This is lower than the literature value for bulk silicon, mainly due to the finite thickness of the membrane. The excitation beam generates electron–hole pairs uniformly in the silicon sample. The transient absorption data presented in Fig. 1d (see also analysis of the decay times in Fig. S2) show that the generated carriers remain for up to tens of nanoseconds, within which period they could have an impact on thermal transport.

**Measurement and data analysis.** Representative TTG signal measured when the excitation pulse is introduced is shown in Fig. 1c (red curve) in comparison with the result without excitation (blue curve). The background signals (obtained with same excitation and probe configuration and without pump) are subtracted to ensure zero baseline. Differences in the signals are clearly seen following the photoexcitation of electron–hole pairs. Following photoexcitation, the rate of decay in the signal is decreased, indicating a slower rate of heat diffusion and thus a reduced thermal conductivity. This is clear evidence of the impact of phonon scattering by free carriers on heat transport. Before the excitation pulse arrives, the two signals are identical, thus ruling out the effect of any accumulated temperature rise from one laser shot to the next due to the excitation pulse at the 1.25-kHz repetition rate of the laser system.

Figure 2a shows TTG signals following the excitation pulse normalized to their values at the excitation pulse arrival time (treated now as time zero) for different excitation pulse energies. The slower decay of the TTG signal as the pulse energy increases indicates the general trend of slower heat diffusion, and accordingly reduced thermal conductivity, at higher carrier densities. The pulse energies in the figure correspond to initially generated carrier densities of ~$2.6 \times 10^{19}$, ~$5.2 \times 10^{19}$, and ~$7.8 \times 10^{19}$ cm$^{-3}$, respectively (see the "Methods" section). At these high carrier concentrations, electron–hole recombination is dominated by Auger processes[32], with timescales in the nanosecond range depending on the carrier density[33]. We have performed a carrier dynamics study by investigating the free carrier absorption of the probe beam under photo-excitation (Fig. 1d). From the transient absorption data, the carrier recombination time is found to be in the nanosecond range (~2–10 ns, see details in Fig. S1), similar to previous results[33]. Such recombination time is sufficiently long that carriers have enough time to scatter phonons. As shown in Fig. 2a, the difference comparing TTG signals with and without excitation is most significant at around 5–10 ns. At later times, the carriers recombine and the thermal conductivity returns to the value of the pristine silicon membrane. As a result, the TTG signal decay rate with photo-excitation also converges to that without excitation. We also note that while the excitation pulse energy was varied by a factor of three, the resulting TTG signals do not exhibit a large difference. This is because at higher carrier

concentrations the recombination time becomes shorter. While the initially generated carrier density is high, the excess carriers recombine faster and do not have sufficient time to interact with phonons. Therefore, increasing the pulse energy further will not lead to appreciable changes in the transient heat transport at nanosecond timescales.

We proceed to extract effective thermal conductivities from the measured TTG signal to quantify the impact of electron–phonon scatterings on heat transport. Because the thermal conductivity varies with time due to continued variations in carrier density, the TTG signal under photo-excitation cannot be fitted using one exponential curve with a single thermal conductivity value (Fig. S3). We denote the time-varying TTG signal (after normalizing to the value at the excitation time, as in Fig. 2a) obtained at an excitation energy $E$ as $I_E(t)$, with the time zero taken to be the excitation time. For time-varying thermal conductivity $k_x(t)$, one can show that the TTG signal decay is given by $I_E(t) = I_E(0)\exp\left[-2\frac{q^2 k_{\mathrm{eff},E}(t)t}{\rho c_p}\right]$, where an effective thermal conductivity is defined as $k_{\mathrm{eff},E}(t) = \frac{1}{t}\int_0^t k_x(\tau)\mathrm{d}\tau$, representing the average thermal conductivity within the time window from zero to $t$ (derivation in the "Methods" section). Without photo-excitation, $k_{\mathrm{eff},E=0}(t)$ simply represents the in-plane thermal conductivity $k_{\mathrm{eff},0}$ of the silicon membrane. To extract the time-varying effective thermal conductivity $k_{\mathrm{eff},E}(t)$ with photo-excitation, it is convenient to analyze the ratio of the TTG signal under photo-excitation to that without excitation: $\bar{I}_E(t) = I_E(t)/I_0(t)$. The decay of this ratio is given by the thermal conductivity difference $k_{\mathrm{eff},E}(t) - k_{\mathrm{eff},0}$, from which $k_{\mathrm{eff},E}(t)$ can be determined based on the silicon membrane's thermal conductivity $k_{\mathrm{eff},0}$ (details are given in the "Methods" section).

The obtained effective thermal conductivity normalized to the thermal conductivity of the silicon membrane $k_{\mathrm{eff},E}(t)/k_{\mathrm{eff},0}$ is shown in Fig. 2b. The normalized thermal conductivities are sensitive to noise and detector bandwidth close to time zero ($t < 1$ ns). Because we focus on the reduction in thermal conductivity, we have limited the upper limit in $y$-axis to 1 (see Fig. S4 for the full curve). Uncertainty in the TTG signals can translate to the extracted thermal conductivity. Owing to the large number of data traces used for the average, we have found the resulting uncertainty in the thermal conductivity is generally <2% (see Fig. S10). For a given excitation pulse energy, the largest thermal conductivity reduction is seen immediately following excitation. Figure 2b clearly demonstrates that the photo-excited electrons and holes have a large impact on the heat transport dynamics— for the highest pulse energy corresponding to an initially generated carrier density of ~7.8 × 10^19 cm^{-3}, the thermal conductivity is reduced by nearly 30%. The reduction in thermal conductivity becomes smaller at later times due to carrier recombination. The TTG signal decay is coupled to the carrier dynamics through the dependence of thermal conductivity on the carrier density. To understand the qualitative features of this coupled phonon–carrier dynamics, a simplified one-dimensional model was used to describe the behavior of the transient thermal decay with free carriers. The recombination lifetimes of electron–hole pairs are taken from the free carrier transient absorption measurements. The thermal conductivity at each carrier density is obtained from first-principles simulations, and the transient TTG signal is obtained from the one-dimensional heat diffusion equation with time-varying thermal conductivity. We then extract the effective thermal conductivity from the simulated TTG signal following the same procedure as discussed above (details of this model are given in the "Methods" section), with the results given in Fig. 2c. The effective thermal conductivities for the highest two pulse energies have only a

small difference, because the faster recombination at higher carrier densities does not allow the initially generated carriers to have sufficient time to scatter phonons. While our model is qualitative as it uses an average recombination lifetime to characterize the carrier dynamics, the general agreement between experiment and the model indicates that the impact of the free carriers on the thermal transport has been captured, mostly because it is the carriers with relatively long recombination lifetimes that significantly scatter phonons.

To understand the cause of the reduction in the thermal conductivity due to electron–phonon interactions, in Fig. 3a we show first-principles-computed phonon scattering rates for each phonon mode due to phonon–electron and phonon–hole interactions at a carrier density of 6 × 10^19 cm^{-3}, in comparison to the intrinsic phonon–phonon scatterings (details of calculation are given in the "Methods" section). At high carrier densities, electrons and holes dominantly scatter low-frequency phonons. This is because the momentum conservation requirement for intravalley electron–phonon scatterings limits participating phonons to those with small wave vectors, and thus low frequencies. Below 3 THz, the phonon scattering rates due to electrons or holes are one order of magnitude larger than the intrinsic phonon–phonon scattering rates. By further combining phonon–phonon and phonon–carrier interactions together with phonon-boundary scatterings, we can calculate the lattice thermal conductivity as a function of carrier density for the silicon membrane (see the "Methods" section). In addition to the lattice contribution, free carriers can also directly contribute to heat transport. These contributions include the electronic thermal conductivity from electrons and holes, as well as the bipolar thermal conductivity[34]. We have also computed these contributions using first principles calculation (details in the "Methods" section), and found in general they are <4% of the lattice thermal conductivity (Fig. S5). Figure 3b shows the computed total thermal conductivity including lattice, electronic, and bipolar contributions. At a carrier concentration of 1 × 10^20 cm^{-3}, the predicted total thermal conductivity is 92 W m^{-1} K^{-1}, which is reduced by 26% from the computed value for the pristine membrane (124 W m^{-1} K^{-1}), highlighting a significant impact of electron–phonon interactions on phonon heat transport. No adjustable parameters have been assumed in the calculation.

To quantify how the thermal conductivity varies with carrier density from the experimental data and facilitate the comparison between theory and experiment, we take the lowest effective thermal conductivity for each given pulse energy from Fig. 2b. The corresponding carrier density $n$ is slightly smaller (by about 20%) than the initially generated carrier density $n_0$ due to carrier recombination, and is taken to be the average value of the carrier density in the time window from 0 to $t$, assuming the carrier density decays exponentially according to $n(t) = n_0\exp(-t/\tau)$, with the decay time $\tau$ given by the free carrier absorption measurement (Fig. S2). Figure 3c shows the thermal conductivities (normalized to that of the silicon membrane without free carriers) at different carrier densities thus obtained from the data for different pulse energies (see Fig. S11 for comparison of the absolute thermal conductivity values without normalization). Each data point in Fig. 3c is an average of at least two locations on the silicon membrane. We have also used a smaller grating period ($L = 5$ μm) and repeated the above experiments (results are given in Fig. S6). For smaller grating periods, the characteristic decay time of heat diffusion is decreased, and becomes comparable to the recombination time (on the order of a few nanoseconds). As a result, the deviation in TTG signal due to photo-excitation appears more pronounced. A one-dimensional heat diffusion model has been used to illustrate the difference in the transient

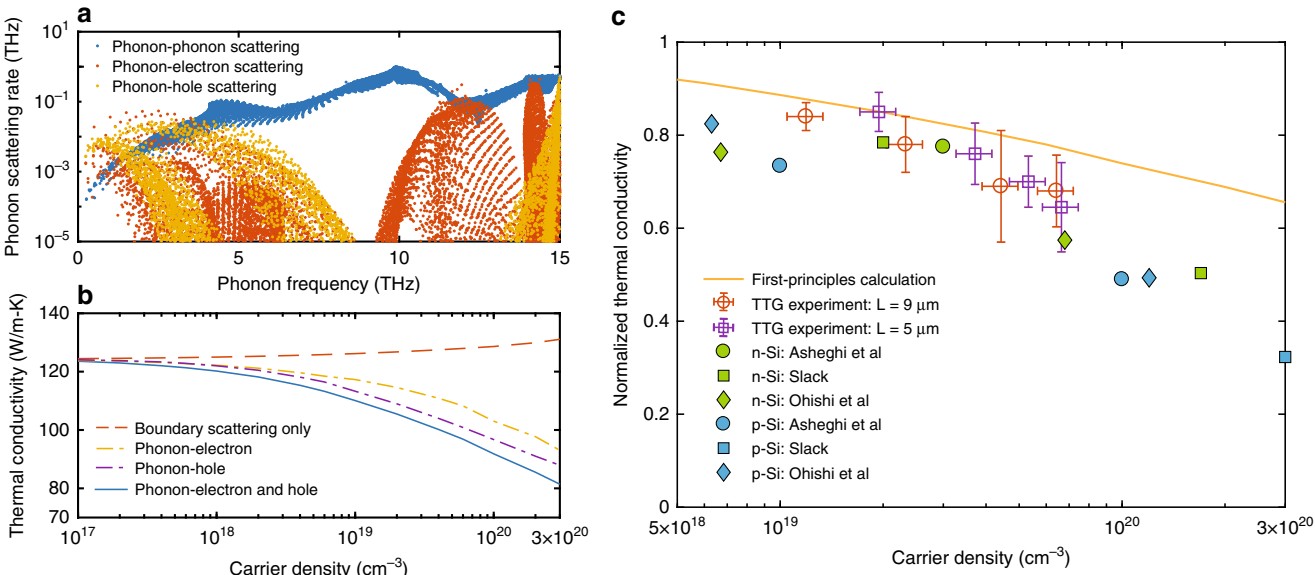

**Fig. 3 Comparison between experiment and first principles calculation. a** First-principles computed phonon scattering rates due to phonon–phonon interactions, phonon–electron interactions, and phonon–hole interactions as functions of the phonon frequency. The carrier density for electrons and holes is taken as $6 \times 10^{19} \, cm^{-3}$. At this carrier concentration, electrons and holes dominantly scatter long wavelength phonons with frequency <3 THz. **b** Computed total thermal conductivity with carrier density including lattice, electronic, and bipolar contributions. The dashed line indicates the total thermal conductivity of the silicon membrane, of which the lattice contribution considers only phonon boundary scatterings and intrinsic phonon–phonon interactions. Other curves further consider the interactions of phonons with electrons, with holes, and with electron–hole pairs together, respectively, when modeling the lattice contribution. At the carrier concentration of $1 \times 10^{20} \, cm^{-3}$, the predicted total thermal conductivity including all scattering mechanisms (blue curve) is 92 W m$^{-1}$ K$^{-1}$, which is reduced by about 25% from the pristine value (124 W m$^{-1}$ K$^{-1}$, corresponding to low carrier density). Details of the calculation are given in the "Methods" section. **c** Normalized thermal conductivity with the carrier density, showing general agreement between experiment and first principles calculation. Each data point is averaged over at least two locations, and we have performed experiments with two different grating periods (5 and 9 µm). The uncertainty represents the standard deviation of the measurement results. The first-principles result is obtained from the solid blue curve in **b** after normalizing to the value for the pristine membrane. Also shown are the thermal conductivities of doped silicon samples with different doping concentrations from past studies.

thermal decay signals depending on the grating period (Fig. S7), which qualitatively agrees with our experiments. In Fig. 3c, the first-principles computed lattice thermal conductivity (normalized to the value of pristine membrane) including phonon–phonon scatterings, phonon-boundary scatterings, and phonon scatterings by electrons and holes has also been included for comparison. In our simulation, electrons and holes are assumed to follow Fermi–Dirac distributions with separate quasi-Fermi levels, which are determined by their densities. This assumption matches with our measurement, because the time-scale of electron–phonon interactions is usually around hundreds of femtoseconds, much faster than the timescale of our experiment. Therefore, we can safely assume that electrons and holes already equilibrate with the lattice and reach their quasi-equilibrium states.

As shown in Fig. 3c, the first principles calculation agrees reasonably well with the experimental results. An alternative way for comparing experiment to theory is to plot the varying effective thermal conductivity from Fig. 2b in terms of the transient carrier density, assuming an exponential decay from the initial density (Fig. S8). While in this approach the carrier dynamics assumes a single average recombination lifetime, the results are consistent with Fig. 3c regarding the reduction in the thermal conductivity. In Fig. 3c we also include the thermal conductivity values measured in silicon samples with different doping concentrations from past studies[23,35,36] (normalized to the bulk value of silicon at room temperature, 148 W m$^{-1}$ K$^{-1}$). In a doped silicon sample (*n*-type or *p*-type), the phonons are scattered by one type of free carriers (electrons or holes), as well as by dopant impurities and

other potential defects introduced by doping. In comparison, the thermal conductivity in our experiments is affected by two types of carriers (both electrons and holes), but without the complica-tion due to dopants or defects. While these two scenarios cannot be directly compared, the similar magnitude of the reduction in thermal conductivity further indicate that electrons and holes can significantly impact phonon heat conduction through the electron–phonon interaction, and suggest that a major proportion in the reduction of the thermal conductivity of conventionally doped silicon may come from phonon–electron scatterings. Furthermore, we note that the magnitude of the reduction in thermal conductivity depends on the sample thickness. Because we used a silicon membrane with a thickness of 2 µm, which is larger than the majority of phonons in silicon at room temperature[37], phonon-boundary scatterings are not dominant, enabling us to uncover the effect due to phonon-carrier scatterings. If a much thinner sample is measured instead, the mean free paths of low-frequency phonons will be strongly limited by boundary scatterings and one would expect a weak carrier density dependence (Fig. S12 illustrates how phonons with different mean free paths are impacted by carrier scattering for different silicon membrane thicknesses).

## Discussion

Here we discuss several issues regarding the analysis and inter-pretation of our measurement. First, the heat generated by the excitation beam will change the thermal response and has to be minimized. The steady-state temperature rise due to the excitation beam is small given the fact that no visible differences in the signals

are observed before the introduction of photo-excitation comparing the cases with and without excitation (Fig. 1c). In Fig. S9, we have also shown that by reducing the repetition rate by half while maintaining the pulse energy (thus decreasing the total excitation power and steady-state heating), the transient heat transport dynamics due to photo-excitation remains the same, further confirming that our measurement is not affected by the steady-state temperature rise. The excitation also creates instantaneous heating due to the thermalization and subsequent recombination of photo-excited carriers. Given the volumetric heat capacity of silicon ($\sim 1.63 \times 10^6$ J K$^{-1}$ m$^{-3}$) and the sample dimension, the maximum instantaneous temperature rise after photo-excitation with the highest pulse energy (6.0 µJ) can be estimated to be ~12 K. As the thermal conductivity of silicon around room temperature approximately follows the $1/T$ trend, we estimate that this temperature change would only lead to a reduction in the lattice thermal conductivity by <4%, which is significantly smaller than the reduction we observed in experiments.

It can be observed from Fig. 3c that, towards the high-carrier concentration, the measured thermal conductivity reduction is slightly greater than the prediction from first principles calculations. There are two possible explanations. First, the measured thermal conductivity of the silicon membrane ($\sim 105$ W m$^{-1}$ K$^{-1}$) is lower than the simulated value for the given membrane thickness, which takes phonon–phonon and phonon–boundary scatterings into consideration ($\sim 124$ W m$^{-1}$ K$^{-1}$). This suggests the sample may contain defects that has further reduced the thermal conductivity. As we have seen, within the acoustic branch for semiconductors, phonon–electron scatterings are most significant for phonons with low frequencies (Fig. 3a). In comparison, phonon-defect scatterings predominantly affect high-frequency phonons[38]. If defects are present, a larger fraction of the total thermal conductivity would be contributed by low-frequency phonons, and therefore we would expect a larger impact of electron–phonon interaction on the thermal transport. In addition, a large number of free carriers result in plasmon excitations—a collective motion of charges. These plasmons, as quasi-particles, can scatter electrons and has been shown to decrease the mobility in the high carrier density region[39]. Similarly, their collisional damping with phonons[40] may also contribute to phonon scatterings, which has not been included in our first principles calculation. Nonetheless, the overall agreement between experiment and simulation indicates that the dominant effect of free carriers on the phonon heat transport below a carrier density of $1 \times 10^{20}$ cm$^{-3}$ is due to electron–phonon interactions.

In summary, we have studied the transient decay of a thermal grating with the presence of photo-excited electrons and holes in a crystalline silicon membrane, and extracted the effective thermal conductivities at different levels of carrier densities. Our results demonstrate that at carrier concentrations above $10^{19}$ cm$^{-3}$, the photo-excited carriers can significantly reduce thermal conductivity by more than 30% on a nanosecond timescale. This is a direct experimental verification of the large impact of electron–phonon interactions on the lattice thermal conductivity at room temperature. The results also open up the prospect of tuning the heat conduction on a nanosecond timescale. The impact of this study is not limited to energy materials such as thermoelectrics. The experimental design can be readily applied to other materials where electron–phonon interactions play an important role such as in phase change materials and molecular crystals, and used to extract important information on electron–phonon coupling. One can further investigate the impact of free carriers on heat conduction along different directions in anisotropic materials (e.g. the cross-plane direction), if heat diffusion into the sample is monitored by means of thermoreflectance method[41–44]. Our technique therefore presents a useful tool to study heat transport in a variety of materials when carriers and phonons are strongly coupled.

## Methods

**Experimental details.** In our TTG set-up, a short-pulsed laser beam at a wavelength of 1030 nm (PHAROS laser, Light Conversion) is split into a TTG pump beam and a carrier excitation beam. A repetition rate of 1.25 kHz is used to minimize steady-state heating. The pump beam is frequency-doubled to 515 nm, while the wavelength of the excitation beam is tuned to 800 nm using an optical parametric amplifier (OPA, ORPHEUS-HP, Light Conversion). The probe beam is provided by a continuous-wave laser (532 nm, Coherent Sapphire SF). Both the pump and probe beams are then diffractively split into two beams with a phase mask, which are subsequently refocused at the sample surface with a cross-angle $\theta$, as shown in Fig. 1a. The excitation beam is focused onto the same area after traveling a longer distance, such that its arrival time is properly delayed relative to the arrival of the pump beam. To minimize beam divergence, the excitation beam is expanded immediately after the OPA. The diffracted probe signal is directed to a fast photodetector (Hamamatsu C5658, 1 GHz bandwidth), and the time-dependent signal is recorded with an oscilloscope (Keysight, MSOS404A, 4 GHz bandwidth). The $1/e^2$ beam sizes of the pump, probe, and excitation beams at the sample plane are measured with a razor-blade beam profiler to be ~200, ~100, and ~280 µm, respectively. The typical pump pulse energy and probe average power before the sample are ~0.7 µJ and ~17 mW, respectively.

The transient thermal measurement is performed on a 2 µm-thick silicon membrane with a lateral dimension of 3.5 mm × 3.5 mm supported on a frame (Norcada). The data are averages of over 100,000 traces for all measurements. At each excitation power, we further acquire background signals by turning on the probe beam only. The background is then subtracted from the TTG signals to ensure zero baseline.

**Estimation of carrier concentration.** The density of electron–hole pairs initially generated by the excitation beam can be estimated via $n_{exc} = \alpha \frac{E_{exc}}{h\nu} \frac{4}{\pi d_{exc}^2 d_{thick}}$, where $\alpha$ is the absorptance of the silicon membrane at 800 nm, $E_{exc}$ is the pulse energy of the excitation, $h\nu$ is the photon energy at 800 nm (1.55 eV), $d_{exc}$ the excitation beam size, and $d_{thick}$ the sample thickness. The absorptance of the sample is measured to be ~40%. For an excitation pulse energy of 6.0 µJ, the above estimation gives an initially generated carrier density of $\sim 7.8 \times 10^{19}$ cm$^{-3}$ for both electrons and holes.

**Determination of effective thermal conductivity.** At zero excitation power, the TTG signal can be well represented by an exponential decay with one single thermal conductivity $k_{eff,0}$: $I_0(t) = I_0(0)\exp[-2\frac{q^2 k_{eff,0} t}{\rho c_p}]$. With finite excitation pulse energy $E$, we expect the TTG signal to decay with time-varying thermal conductivity due to the changing carrier density. As we will show in next section, the variation of TTG signal is governed by the following equation: $\frac{dA}{dt} = -\frac{q^2 k_x(t)}{\rho c_p} A$, where $A$ represents the magnitude of temperature variations. This equation can be integrated to yield $A(t) = A(0)\exp[-q^2[\frac{1}{t}\int_0^t k_x(\tau)d\tau]t/(\rho c_p)]$. Because the TTG signal (light intensity) is proportional to the square of the temperature variation, we have $I_E(t) = I_E(0)\exp[-2q^2[\frac{1}{t}\int_0^t k_x(\tau)d\tau]t/(\rho c_p)]$. Consequently, we define an effective thermal conductivity such that $I_E(t) = I_E(0)\exp[-2q^2 k_{eff,E}(t)t/(\rho c_p)]$, where $k_{eff,E}(t) = \frac{1}{t}\int_0^t k_x(\tau)d\tau$ represents an average thermal conductivity within the time window from zero to $\tau$. To extract the effective thermal conductivity, we first take the ratio of TTG signal at a given pulse energy to that without excitation $\left(\bar{I}_E(t) = \frac{I_E(t)/I_E(0)}{I_0(t)/I_0(0)}\right)$ to minimize the oscillations overlaying on the signal. This time variation of this quantity is expected to be governed by the difference in the thermal conductivity comparing the two cases: $\bar{I}_E(t) = \exp[-2q^2[k_{eff,E}(t) - k_{eff,0}]t/(\rho c_p)]$. The effective thermal conductivity is then evaluated through the following formula (Fig. S4 shows representative data for $\log(\bar{I}_E)$ and corresponding effective thermal conductivities as shown in Fig. 2b).

$$k_{eff,E}(t) = k_{eff,0} - \frac{\rho c_p}{2q^2 t}\log(\bar{I}_E(t)). \tag{1}$$

**Modeling of coupled phonon–carrier dynamics.** To evaluate how the transient thermal decay is impacted by the evolution of carriers due to electron–phonon interaction, we have built a simplified one-dimensional model to capture the qualitative behavior of this coupled phonon–carrier dynamics. The heat diffusion equation assuming one-dimensional thermal grating along $x$ direction with infinite depth is

$$\rho c_p \frac{\partial T}{\partial t} - k_x \frac{\partial^2 T}{\partial x^2} = Q\sin(qx)\delta(t), \tag{2}$$

where $Q$ represents the energy deposited by the pump beam and $\delta(t)$ describes the

short pump pulse at time zero. In practice, because the pump beam has finite penetration depth within the sample and our silicon membrane is relatively thin, the above equation is incorrect due to the missing heat diffusion along $y$ direction (Fig. 1b). However, the quantity that is most relevant to the diffracted signal is the average temperature along the thickness: $\bar{T} = \frac{1}{t} \int_0^t T dz$. Considering the nanoscale timeframe in which we are probing the thermal decay, both boundaries of the silicon membrane can be regarded as adiabatic. It follows that the average temperature $\bar{T}$ still satisfies Eq. (2). Assuming $\bar{T}$ has the form $A(t)\sin(qx)$, Eq. (2) can be solved to give the time variation of the magnitude of average temperature changes

$$\frac{dA}{dt} = -\frac{q^2 k_x(n,p)}{\rho c_p} A, \tag{3}$$

where the dependence of the thermal conductivity on the carrier density has been explicitly written out. We have calculated the thermal conductivity at various electron and hole concentrations (details in the next section). The thermal conductivity at given carrier density is interpolated based on the first-principles calculation results. The carrier dynamics is assumed to be dominated by Auger recombination with a constant Auger recombination lifetime:

$$\begin{cases} \frac{dn}{dt} = -\frac{n}{\tau_{\text{Auger}}} \\ \frac{dp}{dt} = -\frac{p}{\tau_{\text{Auger}}} \end{cases}. \tag{4}$$

We mentioned that surface recombination has been ignored in this model. Silicon generally has small surface recombination velocity and only carrier dynamics at the time scale of around 10 ns or longer is affected by the surface recombination. For example, if we take the surface recombination velocity in silicon to be $S = 10^4$ cm/s, the order-of-magnitude estimation of the corresponding decay time is $\frac{d_{\text{thick}}}{2S} = 10$ ns ($d_{\text{thick}}$ is the sample thickness and the factor of 2 accounts for the two surfaces). Therefore, ignoring the surface recombination will not affect the conclusion of our study which focuses on the initial dynamics. The Auger recombination lifetimes $\tau_{\text{Auger}}$ here are obtained by first fitting the free carrier absorption signal (Fig. 1d) with an exponential decay $A \exp(-t/\tau) + B$, and then interpolating the decay constant data $\tau$ based on the initial carrier density. The fitting range is chosen to be from 0 to 50 ns to represent the average decay rate. The characteristic recombination lifetimes thus obtained, for the initial carrier density from $2.6 \times 10^{19}$, $5.2 \times 10^{19}$ to $7.8 \times 10^{19}$ cm$^{-3}$, are 10.2, 9.1, and 7.2 ns, respectively. Equations (3) and (4) are solved together with the initial condition of $A = 1$ and $n = p = n_{\text{exc}}$. The transient TTG signal is proportional to the square of diffracted beam's electric field, $|\mathbf{E}|^2$, which is proportional to $|A|^2$.

**First principles calculation of thermal conductivity.** We employed the first principles framework described in refs. [37,45,46] to calculate the phonons' and electrons' contributions to the thermal conductivity. For the lattice thermal conductivity contributed by phonons, it is computed given the total phonon relaxation times $\tau_q$ by

$$k_{\text{ph}} = \frac{1}{3VN} \sum_q \hbar \omega_q v_q^2 \tau_q \frac{\partial n_q}{\partial T} \tag{5}$$

where $V$ is the unit cell volume, $N$ is the number of wave vector $\mathbf{q}$ points to be summed up, $\hbar$ the reduced Planck constant, $\omega_q$ the phonon frequency, $\mathbf{v}_q$ the phonon group velocity, and $n_q$ the Bose–Einstein distribution function. The total phonon relaxation times include intrinsic phonon–phonon scatterings, phonon scatterings by carriers, and phonon scatterings by boundaries. The intrinsic phonon–phonon scatterings are determined by anharmonic force constants[37,46] (restricted to third-order force constants in this study). The harmonic and third-order force constants are fitted based on first principles data of force-atom displacement in a supercell for different sets of displacements (a supercell with $2 \times 2 \times 2$ conventional unit cells and 64 atoms is used). The harmonic force constants determine the phonon dispersion and the third-order force constants are further used to calculate the phonon–phonon relaxation times ($\tau_{\text{ph-ph}}$). We have also checked thermal conductivity results obtained with $4 \times 4 \times 4$ primitive cell (128 atoms) for third-order force constant calculation and the resulting thermal conductivity differs by <5%, indicating that the results obtained with $2 \times 2 \times 2$ conventional unit cells are converged.

The phonon scattering rates by carriers depend on the electron–phonon interactions, which are first obtained using density functional perturbation theory as implemented in the QUANTUM ESPRESSO[47] package on a coarse mesh for electrons ($12 \times 12 \times 12$) and phonons ($6 \times 6 \times 6$). We then use the EPW package[48] with the Wannier interpolation[49] scheme to map the electron–phonon interaction information to a much denser mesh ($60 \times 60 \times 60$ for both electrons and phonons) and evaluate all possible scattering channels for each phonon over the Brillouin zone based on Fermi's Golden rule. The electrons and holes are assumed to follow Fermi–Dirac distribution function with quasi-Fermi levels given by their respective number densities. The phonon scattering rates due to free carriers sum up the contributions from both electrons ($\tau_{\text{ph-e}}$) and holes ($\tau_{\text{ph-h}}$). The total phonon relaxation times without considering the boundary scatterings are given by Matthiessen's rule: $\frac{1}{\tau_{q,\text{bulk}}} = \frac{1}{\tau_{\text{ph-ph}}} + \frac{1}{\tau_{\text{ph-e}}} + \frac{1}{\tau_{\text{ph-h}}}$.

The phonon-boundary scattering is treated within the framework of Fuchs–Sondheimer model[50,51]. The interaction between phonons and boundary in the thin silicon film effectively modifies the relaxation time by the following reduction factor (details of the derivation is given in Supplementary Note 1)

$$S_q = 1 - \frac{\tau_{q,\text{bulk}} |v_z|}{t} \frac{(1 - p_s)\left(1 - e^{-\frac{t}{\tau_{q,\text{bulk}} |v_z|}}\right)}{1 - p_s e^{-\frac{t}{\tau_{q,\text{bulk}} |v_z|}}}, \tag{6}$$

where $v_z$ is the $z$-component of phonon group velocity, and $p_s$ is an average specularity ratio that characterizes how much fraction of phonons impinging on the boundary are reflected specularly. The remaining portion is assumed to be diffusely scattered at the boundaries. In general, long wavelength phonons are less sensitive to the surface roughness and are more specularly scattered, while small wavelength phonons are diffusely scattered. For our simulation we have taken $p_s = 0.0$, essentially assuming fully diffuse boundary scattering. As is clear from the reduction factor, phonons traveling parallel with the film are less affected by the boundary scatterings. The total phonon relaxation times in the silicon membrane are given by $\tau_{q,\text{film}} = S_q \tau_{q,\text{bulk}}$, which is used to calculate the lattice thermal conductivity based on Eq. (5). The resulting thermal conductivities at different carrier densities, compared to that without considering phonon–electron scatterings, are given in Fig. 3b. These thermal conductivity values normalized to the pristine value of silicon membrane give the curve shown in Fig. 3c.

The electronic contributions to the thermal conductivity from electrons can be estimated by[45]

$$k_e = \frac{1}{3VNT} \sum_k (E - \mu)^2 v_k^2 \tau_k \left(-\frac{\partial f_k}{\partial E}\right) - T\sigma_e S_e^2, \tag{7}$$

conduction band

where the summation includes the conduction band, $\mu$ is the quasi-Fermi level for electrons, $v_k$ the electron group velocity, $\tau_k$ the electron relaxation time, $f_k$ the Fermi–Dirac distribution function, $\sigma_e$ electrical conductivity, and $S_e$ the Seebeck coefficient. A similar expression that considers the valence band with a quasi-Fermi level for holes gives holes' contribution to the thermal conductivity ($k_h$). The relaxation times for electrons and holes are dominated by electron–phonon scatterings, which are computed using the EPW package similarly as for phonon–electron scatterings. When both electrons and holes are present, there is also an additional contribution to the thermal conductivity which arises from the generation and recombination of electron–hole pairs in different spatial locations, known as bipolar thermal conductivity. This contribution is given by $k_{\text{bipolar}} = T\frac{\sigma_e \sigma_h}{\sigma_e + \sigma_h}(S_e - S_h)^2$. The bipolar thermal conductivity is derived under open-circuit condition assuming electrons and holes have same local quasi-Fermi levels, which is not strictly valid in our experiment. Nonetheless, the number should suggest the order of magnitude of the free carriers' contribution to the thermal conductivity. The computed total contribution from electrons and holes including the bipolar contribution is shown in Fig. S5. Up to $1 \times 10^{20}$ cm$^{-3}$, this total contribution only comprises a small fraction (<4%) of the lattice thermal conductivity.

## Data availability

The data that support the findings of this study are available from the corresponding author on reasonable request.

## Code availability

The code for computing phonon–electron scattering rates through first principles electron transport calculation is a modified version of the EPW code[48], originally released within the QUANTUM ESPRESSO package[47]. Our modified EPW code is available at https://doi.org/10.24435/materialscloud:5a-7s.

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

## Acknowledgements

We thank J. Shin and B. Liao for helpful discussions on the experimental set-up. This work is supported by the DARPA MATRIX program, under Grant No. HR0011-16-2-0041.

## Author contributions

J.Z., G.C. conceived the idea. J.Z., H.D.S. carried out the experiment, with help from K.C., B.S., R.A.D., and A.A.M. J.Z., Q.X. performed the first principles calculations. J.Z. and G.C. analyzed the data and wrote the manuscript, with contribution from A.A.M. and K.A.N. G.C. supervised the research. All authors commented on, discussed, and edited the manuscript.

## Competing interests

The authors declare no competing interests.
