## [Peer Review File · Nature Communications]

REVIEWER COMMENTS

Reviewer #1 (Remarks to the Author):

The authors have dealt with an important issue in the present manuscript: the effect of electron-phonon interaction on phonon transport at room temperature. The electron-phonon interaction is known to have relatively dominant effect on the phonon transport at lower temperature where magnitude of phonon-phonon scattering is low. The authors have devised a novel experimental technique to study the impact of electron-phonon scattering on the phonon transport at room temperature using modified transient thermal grating technique and supplemented the experimental observations with theoretical modelling. The authors have found that electron-phonon interaction could reduce thermal conductivity by as large as 30% for carrier concentration $\sim 8 \times 10^{19} \text{ cm}^{-3}$ in silicon membrane. The conclusions are well supported by experimental data and the newly devised technique could be very useful in studying correlated electron-phonon dynamics. I recommend publication of this manuscript. I have few minor concerns which are listed below:

1. More details should be given about data processing. For example, data in Fig. 2a is raw data or post-processed? Many ripples can be seen which appears to be random in delay time but are highly correlated among different photo-excitation energy. Such ripples can be seen in the derived thermal conductivity shown in Fig. 2b as well. Are these ripples related to any electron-phonon dynamics or just measurement artifact? Please explain.
2. Difference in signal among various excitation fluences in Fig. 2a seems to be very small – what is the error limit and confidence level? Please show all these data together with their respective fitted curve with error bar/confidence level in a separate single graph. It is important because while the signals shown in Fig. 2a for various excitation fluences have very small difference from one another, the derived thermal conductivity shown in Fig. 2b seems to be affected strongly.
3. Data is shown for two different repetition rate - 1.25 kHz and 0.625 kHz (supplementary Fig. 9) to check steady state heating. While the authors claim these two data set overlaps with each other, I can see that difference among these two data sets of same order as different excitation fluence shown in Fig. 2a. Is it possible that the effect of high repetition rate is similar to excitation fluence as explained in page 7 of the manuscript, and the difference among signal decreases because of faster recombination rate? Can we see data for lower repetition rate?
4. The difference among relative ratio of TTG signal for different excitation fluences (supplementary Fig.4a) increases with increasing time. This seems to be contradictory as the effect of electron-phonon interaction decreases with increasing time. The signal in Fig.2a and the derived thermal conductivity (supplementary Fig. 4b), however, seems to be correct. Please explain.
5. Please indicate value of the exponent in supplementary Fig. 3 in their corresponding curve.
6. I believe the caption of supplementary Figure 1 should be TTG signal without photo-excitation.

Reviewer #2 (Remarks to the Author):

The authors studied the electron-phonon interaction effect on phonon heat transport, using silicon membrane as sample and employing photoexcitation technique, transient heat transport measurement as well as the state-of-art first-principles calculation. The carriers are successfully modulated by photoexcitation and then the electron-phonon interaction and also the thermal conductivity. The present work shows potential way to analyze the electron-phonon interaction quantitatively in real systems.

I generally recommend its publication in Nature Communications. However, the following issues need to be addressed:

- 1) In calculations of phonon scattering rates using QUANTUM-ESPRESSO and EPW, the coarse mesh and the denser mesh should be specified. Besides, the $2 \times 2 \times 2$ conventional unit cells, 96 atoms, for third order force constants calculation are somewhat small. The convergence should be tested.
- 2) In Fig. 3, I was expecting to see a figure showing direct comparison of experimental and theoretical thermal conductivity induced by electron-phonon interaction.

Reviewer #3 (Remarks to the Author):

The authors provide experimental and numerical evidence that the enhanced phonon scattering by photoexcited free carriers results in a substantial reduction in thermal conductivity. The authors claim that they provide direct experimental evidence of the role of electron-phonon interaction in phonon heat transport, which is important for understanding heat conduction in doped semiconductors. The authors generated spatially periodic temperature modulation and monitored the decay of the thermal grating with introducing excited electron-hole pairs. They found that for an initially generated carrier density of $\sim 7.8 \times 10^{19} \text{ cm}^{-3}$, the thermal conductivity is reduced by nearly 30%. To understand the mechanism of reducing the thermal conductivity makes new finding a breakthrough in the field of designing good thermoelectric materials.

I have the several comments.

1. The authors provided the theoretical carrier density dependence of thermal conductivity in Fig. 3 (b). My question is, how sensitive (in experimental), are the boundary scattering contribution to the thermal conductivity. In Fig. 3 (c), the experimental carrier density dependence of thermal conductivity are in good agreement with theoretical result. However, if the boundary scattering is dominant in thinner film or small sample, experimental carrier density dependence might be different.
2. The authors should provide sufficient information on theoretical carrier density dependence of electronic part of thermal conductivity. The electronic part of thermal conductivity should increase if the carrier density increase.
3. Is it possible to evaluate the reduction of electrical conductivity by electron-phonon interaction effect experimentally?

Revision report for MS# NCOMMS-20-31595

Direct observation of large electron-phonon interaction effect on phonon heat transport

Jiawei Zhou, Hyun D. Shin, Ke Chen, Bai Song, Ryan A. Duncan, Qian Xu, Alexei A. Maznev, Keith A. Nelson, Gang Chen

We thank all the reviewers for their thoughtful comments on the manuscript. Our responses and revisions (in blue) are elaborated on below. In addition, the revised parts in the main manuscript and the supplementary information are also highlighted (in yellow). The revised manuscript and supplementary information are attached in this document, following our revision report.

Reviewer #1:

The authors have dealt with an important issue in the present manuscript: the effect of electron-phonon interaction on phonon transport at room temperature. The electron-phonon interaction is known to have relatively dominant effect on the phonon transport at lower temperature where magnitude of phonon-phonon scattering is low. The authors have devised a novel experimental technique to study the impact of electron-phonon scattering on the phonon transport at room temperature using modified transient thermal grating technique and supplemented the experimental observations with theoretical modelling. The authors have found that electron-phonon interaction could reduce thermal conductivity by as large as 30% for carrier concentration $\sim 8 \times 10^{19} \text{ cm}^{-3}$ in silicon membrane. The conclusions are well supported by experimental data and the newly devised technique could be very useful in studying correlated electron-phonon dynamics. I recommend publication of this manuscript. I have few minor concerns which are listed below:

Response: We are glad the reviewer finds the experimental technique novel and we thank the reviewer for his/her support for publication. One of the reviewer's main comments is about data analysis, including the details of data processing, and the confidence interval of signals. The reviewer also raised a question about the effect of repetition rate. We have provided additional data analysis to address these comments, and appreciate the reviewer's suggestions that have helped to improve the manuscript. Our detailed responses can be found below.

1. More details should be given about data processing. For example, data in Fig. 2a is raw data or post-processed? Many ripples can be seen which appears to be random in delay time but are highly correlated among different photo-excitation energy. Such ripples can be seen in the derived thermal conductivity shown in Fig. 2b as well. Are these ripples related to any electron-phonon dynamics or just measurement artifact? Please explain.

Response: We thank the reviewer for the suggestion on clarifying the data processing. Data presented in Fig. 2a are signals normalized to their values at the time of excitation, after subtracting baseline signals (obtained with same excitation and probe power but pump is turned off). We suspect the ripples are due to electronic noise (these ripples exist when no photo-excitation is used). For the measurement, we have selected impedance matched resistance when reading signals from the oscilloscope and added radio frequency choke on the cable, which have reduced the electronic noises. We have added clarification about data processing in the manuscript.

Revision: In the *Measurement and data analysis* section of the text, and figure caption of Fig. 1, we added

“The background signals (obtained with same excitation and probe configuration and without pump) are subtracted to ensure zero baseline.”

In the figure caption of Fig. 2, we changed the first sentence to

“Signals (after subtracting the background signals) under different excitation fluences normalized to their values at the excitation time, which is taken as the time zero.”

2. Difference in signal among various excitation fluences in Fig. 2a seems to be very small – what is the error limit and confidence level? Please show all these data together with their respective fitted curve with error bar/confidence level in a separate single graph. It is important because while the signals shown in Fig. 2a for various excitation fluences have very small difference from one another, the derived thermal conductivity shown in Fig. 2b seems to be affected strongly.

Response: We thank the reviewer for the suggestion to include confidence intervals in the figure. In Fig. R1 below, we plot the 95% confidence intervals of TTG signals and corresponding extracted effective thermal conductivities at different excitation fluences. The results shown in Fig. 2a are averaged over 100,000 traces of measurements (see *Experimental details* section in Methods, page 14), and at each time point we can calculate the 95% confidence interval by evaluating the standard error of the sample mean. The variation of the 95% confidence interval with time then forms shaded regions, as shown in Fig. R1a for different excitation fluences. While the difference in signal among various excitation fluences are small, the standard error is even smaller which enables us to distinguish different curves.

The standard error in the TTG signals translate to standard error in the extracted effective thermal conductivity. In Fig. R2b we show the resulting 95% confidence interval for the effective thermal conductivities at different excitation fluences, corresponding to different initial carrier densities. Owing to the small standard error in TTG signals, the uncertainty in the extracted thermal conductivity for a single measurement is generally less than 2%.

Figure R1. Confidence interval of TTG signals and extracted effective thermal conductivities at different excitation fluences.

Revision: We have included the confidence interval plot (Fig. R1) as a separate figure in the Supplementary Information (Fig. S10), and have added the following text in the manuscript (page 8), referring to this figure:

“Uncertainty in the TTG signals can translate to the extracted thermal conductivity. Owing to the large number of data traces used for the average, we have found the resulting uncertainty in the thermal conductivity is generally less than 2% (see Fig. S10).”

3. Data is shown for two different repetition rate - 1.25 kHz and 0.625 kHz (supplementary Fig. 9) to check steady state heating. While the authors claim these two data set overlaps with each other, I can see that difference among these two data sets of same order as different excitation fluence shown in Fig. 2a. Is it possible that the effect of high repetition rate is similar to excitation fluence as explained in page 7 of the manuscript, and the difference among signal decreases because of faster recombination rate? Can we see data for lower repetition rate?

Response: The difference between the two data sets at different repetition rates shown in Fig. S9 is mostly due to the laser power drift. The small variation in the laser power creates variation in the TTG signal, and can be seen before the excitation pulse arrives. If we subtract the baseline signal (signal before the pump arrives) and normalize all signals to a fixed time point (here we choose $t = 5$ ns), the resulting data are shown in Fig. R2. It can be seen that the difference between the two data sets at different repetition rates is negligible.

In our experiment, the repetition rate is changed keeping the excitation pulse energy the same. Therefore, the effect of increasing the repetition rate is mainly to increase the average temperature of the silicon membrane sample due to steady state heating, and is different from the effect of increasing excitation fluence (at fixed repetition rate). This is because at a repetition rate of 1.25 kHz, the excitation pulses are separated by 800 μ s, sufficiently longer than typical carrier recombination time in silicon¹ such that most carriers have recombined before the next excitation pulse arrives.

Figure R2. TTG signals under photo-excitation with two different repetition rates (1.25 kHz and 0.625 kHz) compared to the TTG signal without photo-excitation. The baseline signals before the pump arrives are subtracted to ensure zero baseline, and the signals are normalized to their values at $t = 5$ ns.

Revision: We have added the following sentences in the figure caption of Fig. S9 in the Supplementary Information to explain the difference between signals at different repetition rates:

“The difference between the signals at different repetition rates is mostly due to the laser power drift. The small variation in the laser power creates variation in the TTG signal, and can be seen before the excitation pulse arrives.”

4. The difference among relative ratio of TTG signal for different excitation fluences (supplementary Fig.4a) increases with increasing time. This seems to be contradictory as the effect of electron-phonon interaction decreases with increasing time. The signal in Fig.2a and the derived thermal conductivity (supplementary Fig. 4b), however, seems to be correct. Please explain.

Response: We think the results in supplementary Fig. 4a are not in contradiction with the fact that the electron-phonon scattering effect diminishes with time. The logarithmic relative ratio of TTG signals shown in Fig. S4a increases with increasing time because the plotted quantity:

$$\log (\bar{I}_E) = \log \left(\frac{I_E(t)/I_E(0)}{I_0(t)/I_0(0)} \right)$$

represents the cumulative effect of electron-phonon interaction on the thermal decay from the time of excitation. This quantity is directly related to the effective thermal conductivity within the time window from zero to t , as shown in Methods (section *Determination of effective thermal conductivity*):

$$\log (\bar{I}_E) = 2q^2 [k_{eff,0} - k_{eff,E}(t)]t/(\rho c_p)$$

With the definition of $k_{eff,E}(t) = \frac{1}{t} \int_0^t k_x(\tau) d\tau$, we can show that

$$\log (\bar{I}_E) = \frac{2q^2}{\rho c_p} \int_0^t [k_{eff,0} - k_x(\tau)] d\tau$$

where $k_x(t)$ is the instantaneous thermal conductivity at time t . At later times, when the effect of electron-phonon interaction diminishes and the thermal conductivity approaches the intrinsic value, the quantity $\log (\bar{I}_E)$ approaches a constant, in agreement with Fig. S4a. Therefore we think the results presented in Fig. S4a are not in contradiction with the diminishing electron-phonon interaction effect on phonon heat transport with increasing time.

Revision: In the figure caption of Fig. S4, we added

“The quantity $\log (\bar{I}_E)$ represents the cumulative effect of electron-phonon interaction on the thermal decay from the time of excitation. With the definition of the effective thermal conductivity given in Methods, we can show that $\log (\bar{I}_E) = \frac{2q^2}{\rho c_p} \int_0^t [k_{eff,0} - k_x(\tau)] d\tau$, where $k_x(t)$ is the instantaneous thermal conductivity at time t . At later times when the effect of electron-phonon interaction diminishes, the quantity $\log (\bar{I}_E)$ approaches a constant, as shown in (a).”

5. Please indicate value of the exponent in supplementary Fig. 3 in their corresponding curve.

Response: We thank the reviewer for the suggestion and have added the exponent in Fig. S3 caption.

Revision: In the figure caption of Fig. S3 we added the following information

“Curves with different values of b are shown as dashed lines ($b = 0.060 \text{ ns}^{-1}, 0.051 \text{ ns}^{-1}, 0.042 \text{ ns}^{-1}$ for curves from the lowest to the highest, respectively, corresponding to thermal conductivity values of $k = 100, 85, 70 \text{ W/mK}$ at a grating period of $9 \mu\text{m}$) ...”

6. I believe the caption of supplementary Figure 1 should be TTG signal without photo-excitation.

Response: We thank the reviewer for pointing this out. We have corrected this in the revised Supplementary Information.

Revision: We changed the first sentence in the figure caption of Fig. S1 to

“Transient thermal grating signal without photo-excitation.”

Reviewer #2:

The authors studied the electron-phonon interaction effect on phonon heat transport, using silicon membrane as sample and employing photoexcitation technique, transient heat transport measurement as well as the state-of-art first-principles calculation. The carriers are successfully modulated by photoexcitation and then the electron-phonon interaction and also the thermal conductivity. The present work shows potential way to analyze the electron-phonon interaction quantitatively in real systems.

I generally recommend its publication in Nature Communications. However, the following issues need to be addressed:

Response: We thank the reviewer for his/her positive comments and his/her suggestions that helped to improve the manuscript. The reviewer's major comments are about the convergence of thermal transport calculation, and comparison between experiment and simulation. We have provided additional information to address these comments. Below are our detailed responses.

1) In calculations of phonon scattering rates using QUANTUM-ESPRESSO and EPW, the coarse mesh and the denser mesh should be specified. Besides, the $2 \times 2 \times 2$ conventional unit cells, 96 atoms, for third order force constants calculation are some what small. The convergence should be tested.

Response: We thank the reviewer for the suggestion. For computing phonon scattering rates by carriers, the coarse meshes are $12 \times 12 \times 12$ for electrons and $12 \times 12 \times 12$ for phonons. The denser meshes are $60 \times 60 \times 60$ for both electrons and phonons. These have been specified in the revised manuscript.

We have also checked thermal conductivity results obtained with $4 \times 4 \times 4$ primitive cell (128 atoms) for third order force constant calculation and the resulting thermal conductivity differs by less than 5%. This small difference indicates the results obtained with $2 \times 2 \times 2$ conventional unit cells are converged.

Revision: We have added the mesh information in Methods (section *First principles calculation of thermal conductivity*)

“The phonon scattering rates by carriers depend on the electron-phonon interactions, which are first obtained using density functional perturbation theory as implemented in the QUANTUM ESPRESSO² package on a coarse mesh for electrons ($12 \times 12 \times 12$) and phonons ($6 \times 6 \times 6$). We then use the EPW package³ with the Wannier interpolation⁴ scheme to map the electron-phonon interaction information to a much denser mesh ($60 \times 60 \times 60$ for both electrons and phonons) ...”

and added the following sentences regarding the convergence test

“We have also checked thermal conductivity results obtained with $4 \times 4 \times 4$ primitive cell (128 atoms) for third order force constant calculation and the resulting thermal conductivity differs by

less than 5%, indicating that the results obtained with $2 \times 2 \times 2$ conventional unit cells are converged.”

We have also corrected a typo in Methods (the number of atoms in $2 \times 2 \times 2$ conventional unit cells should be 64):

“The harmonic and third-order force constants are fitted together based on first principles calculations in a supercell that give forces acting on different atoms for different sets of displacements ($2 \times 2 \times 2$ conventional unit cells, 64 atoms).”

2) In Fig. 3, I was expecting to see figure showing direct comparison of experimental and theoretical thermal conductivity induced by electron-phonon interaction.

Response: In Fig. 3c, we plot normalized thermal conductivity to show the relative changes induced by electron-phonon interactions. In Fig. R3 below, we show the direct comparison between the experimental and theoretical thermal conductivities. In this plot, the calculation of the thermal conductivity also considers the size effect due to the finite grating period, using an approximate solution based on the variational approach⁵ (see Supplementary Note 2 we added in the revised Supplementary Information). This solution is used because rigorous solution for the thin film geometry with finite grating period would require expensive numerical simulations.

In Fig. R3, the measured thermal conductivities of the silicon membrane without photo-excitation for both grating periods ($9 \mu\text{m}$ and $5 \mu\text{m}$) are lower than the simulated values. This is possibly because the sample contains defects that have further reduced the thermal conductivity. The measured thermal conductivities at $5 \mu\text{m}$ is lower than those at $9 \mu\text{m}$ because the smaller grating period imposes a stronger size effect on the thermal transport. Nonetheless, the decreasing trend in the thermal conductivity with increasing carrier density agree between experiment and simulation. We have added this plot in the Supplementary Information.

Figure R3. Direct comparison between experimental and theoretical thermal conductivity.

Revision: We have included Fig. R3 in the Supplementary Information as Fig. S11, with figure caption

“Direct comparison between experimentally extracted effective thermal conductivities at different grating periods and first-principles-computed thermal conductivity with respect to the carrier density. The calculation of the thermal conductivity in this figure also considers the size effect due to the finite grating period, using an approximate solution based on the variational approach (see Supplementary Note 2). The experimental thermal conductivities lie below the theoretical curve possibly because the sample contains defects that have further reduced the thermal conductivity. The thermal conductivities obtained with a grating period of 5 μm are smaller than those at 9 μm because the smaller grating period imposes a stronger size effect on the heat transport. Nonetheless, the decreasing trend in the thermal conductivity as the carrier density increases agree between experiment and simulation.”

and added Supplementary Note 2 to explain the computation of thermal conductivity of a thin film silicon considering the size effect due to finite grating period.

We also modified the following sentence in the manuscript (page 10), referring to Fig. S11:

“Figure 3c shows the thermal conductivities (normalized to that of the silicon membrane without free carriers) at different carrier densities thus obtained from the data for different pulse energies (see Fig. S11 for comparison of the absolute thermal conductivity values without normalization).”

Reviewer #3:

The authors provide experimental and numerical evidence that the enhanced phonon scattering by photoexcited free carriers results in a substantial reduction in thermal conductivity. The authors claim that they provide direct experimental evidence of the role of electron-phonon interaction in phonon heat transport, which is important for understanding heat conduction in doped semiconductors. The authors generated spatially periodic temperature modulation and monitored the decay of the thermal grating with introducing excited electron-hole pairs. They found that for an initially generated carrier density of $\sim 7.8 \times 10^{19} \text{ cm}^{-3}$, the thermal conductivity is reduced by nearly 30%. To understand the mechanism of reducing the thermal conductivity makes new finding a breakthrough in the field of designing good thermoelectric materials. I have the several comments.

Response: We thank the reviewer for his/her positive comments on our work, and the suggestions that helped to improve our manuscript. The reviewer's main comments are on the effect of boundary scattering, and the electronic contribution to the thermal conductivity. We have added discussions in the manuscript to address these comments. Below are our detailed responses.

1. The authors provided the theoretical carrier density dependence of thermal conductivity in Fig. 3 (b). My question is, how sensitive (in experimental), are the boundary scattering contribution to the thermal conductivity. In Fig. 3 (c), the experimental carrier density dependence of thermal conductivity are in good agreement with theoretical result. However, if the boundary scattering is dominant in thinner film or small sample, experimental carrier density dependence might be different.

Response: We agree with the reviewer that if the boundary scattering becomes dominant, the carrier density dependence of the thermal conductivity will be different. When comparing to the experimental thermal conductivity, we have considered the phonon boundary scattering in our simulations. Because the silicon membrane we used has a thickness of 2 μm , which is larger than the majority of phonons in silicon at room temperature⁶, phonon-boundary scatterings are not dominant in this case.

If a thinner film is measured, because the boundary scattering dominantly affects long mean free path (low frequency) phonons, which are also the phonons that are mostly affected by carrier scatterings (Fig. 3a), one would expect weaker carrier density dependence. To illustrate this effect, we computed accumulated thermal conductivity (including both lattice and electronic contributions as in Fig. 3b) with respect to phonon mean free path at different carrier densities for bulk silicon and silicon films with different thicknesses, as shown in Fig. R4 below. Carrier density of 10^{17} cm^{-3} is close to the intrinsic case with phonon mean free paths spanning from nanometer to micrometer. For a bulk silicon sample at the carrier density of 10^{19} cm^{-3} , carriers dominantly scatter phonons with mean free paths longer than around 2 μm , leading to a thermal conductivity reduction of about 20% (Fig. R4a). At 10^{20} cm^{-3} , carriers mainly scatter phonons with mean free paths longer than around 1 μm , with nearly 40% reduction in thermal conductivity (Fig. R4a).

For a silicon thin film with 2 μm thickness, phonons have reduced mean free paths due to boundary scattering. Nonetheless, phonons with mean free paths longer than 1 μm still contribute

substantially (about 25%) to thermal conductivity (Fig. R4b). Therefore, increasing carrier density will further scatter these phonons and reduce the thermal conductivity. The thermal conductivity reduction at carrier densities of 10^{19} cm^{-3} and 10^{20} cm^{-3} is about 10% and 30% respectively.

If the silicon film thickness is further reduced to 500 nm, phonons are now strongly scattered by boundaries, and it can be seen that phonons with mean free paths longer than $1 \mu\text{m}$ contribute to only about 20% of total thermal conductivity (Fig. R4c). In such case, the effect of increasing the carrier density on the thermal conductivity is smaller. The thermal conductivity reduction at the carrier density of 10^{19} cm^{-3} and 10^{20} cm^{-3} is about 7% and 20% respectively (Fig. R4c). Further decreasing the film thickness will make the carrier scattering effect even smaller.

To illustrate how phonon boundary scattering affects the observation of carrier density dependence of thermal conductivity, we have added discussion in the manuscript regarding the effect of phonon-boundary scattering, and added Fig. R4 in the Supplementary Information.

Figure R4. Computed accumulated thermal conductivity with respect to the phonon mean free path at different carrier densities, for (a) bulk silicon, (b) silicon membrane with 2 μm thickness, and (c) silicon membrane with 500 nm thickness.

Revision: In the discussion following Fig. 3 (page 11) we have added the following sentences

“Furthermore, we note that the magnitude of the reduction in thermal conductivity depends on the sample thickness. Because we used a silicon membrane with a thickness of 2 μm , which is larger than the majority of phonons in silicon at room temperature⁶, phonon-boundary scatterings are not dominant, enabling us to uncover the effect due to phonon-carrier scatterings. If a much thinner sample is measured instead, the mean free paths of low frequency phonons will be strongly limited by boundary scatterings and one would expect a weak carrier density dependence (Fig. S12 illustrates how phonons with different mean free paths are impacted by carrier scattering for different silicon membrane thicknesses).”

and added Fig. R4 in the Supplementary Information as Fig. S12 with the figure caption:

“Computed accumulated thermal conductivity with respect to the phonon mean free path at different carrier densities, for (a) bulk silicon, (b) silicon membrane with 2 μm thickness, and (c) silicon membrane with 500 nm thickness. The thermal conductivity includes both lattice and electronic contributions as in Fig. 3b. Carrier density of 10^{17} cm^{-3} is close to the intrinsic case with phonon mean free paths spanning from nanometer to micrometer. For a bulk silicon sample at the carrier density of 10^{19} cm^{-3} , carriers dominantly scatter phonons with mean free paths longer than around 2 μm , leading to a thermal conductivity reduction of about 20% (a). At 10^{20} cm^{-3} , carriers dominantly affect phonons with mean free paths longer than around 1 μm , with nearly 40% reduction in thermal conductivity (a). For a silicon membrane with 2 μm thickness, phonons with mean free paths longer than 1 μm still contribute substantially (about 25%) to the thermal conductivity. As a result, the thermal conductivity reduction at carrier densities of 10^{19} cm^{-3} and 10^{20} cm^{-3} is about 10% and 30% respectively. If the silicon film thickness is further reduced to 500 nm, phonons are now strongly scattered by boundaries. In such case, the effect of increasing the carrier density is smaller. The thermal conductivity reduction at the carrier density of 10^{19} cm^{-3} and 10^{20} cm^{-3} is only about 7% and 20% respectively.”

2. The authors should provide sufficient information on theoretical carrier density dependence of electronic part of thermal conductivity. The electronic part of thermal conductivity should increase if the carrier density increase.

Response: We agree with the reviewer that the electronic contribution to the thermal conductivity is important, and therefore have included the theoretical carrier density dependence of the electronic part of thermal conductivity in supplementary Figure 5 (including contributions from both electrons and holes, and bipolar contribution). As pointed out by the reviewer, the electronic part of thermal conductivity increases with increasing carrier density (Fig. S5). We found that at a carrier density of $1 \times 10^{20} \text{ cm}^{-3}$ (for both electrons and holes), the total contributions from electrons and holes to the thermal conductivity is $\sim 4.5 \text{ W/mK}$. In Fig. 3b, the presented thermal conductivities have included such electronic contributions (the increasing thermal conductivity with increasing carrier density without phonon-carrier scattering is due to the electronic contributions, red dashed line in Fig. 3b).

3. Is it possible to evaluate the reduction of electrical conductivity by electron-phonon interaction effect experimentally?

Response: Different from the heat conduction case where electron-phonon interaction is usually believed to be a minor effect, for electron conduction, electron-phonon interaction is often the dominant scattering process and usually determines the electrical conductivity of semiconductors⁷. Regarding the possibility of evaluating the interactions between electrons and specific phonons, we think experimental techniques which selectively excite phonon modes while detecting electron conduction via e.g. conductance measurement could be a viable approach. We believe this is an interesting direction to explore. However, because this is beyond the scope of current work, we do not discuss this in the manuscript.

Reference

1. Dziwior, J. & Schmid, W. Auger coefficients for highly doped and highly excited silicon. *Appl. Phys. Lett.* **31**, 346–348 (1977).
2. Giannozzi, P. *et al.* QUANTUM ESPRESSO: a modular and open-source software project for quantum simulations of materials. *J. Phys.: Condens. Matter* **21**, 395502 (2009).
3. Giustino, F., Cohen, M. & Louie, S. Electron-phonon interaction using Wannier functions. *Phys. Rev. B* **76**, 165108 (2007).
4. Marzari, N., Mostofi, A. A., Yates, J. R., Souza, I. & Vanderbilt, D. Maximally localized Wannier functions: Theory and applications. *Rev. Mod. Phys.* **84**, 1419–1475 (2012).
5. Chiloyan, V. *et al.* Variational approach to solving the spectral Boltzmann transport equation in transient thermal grating for thin films. *Journal of Applied Physics* **120**, 025103 (2016).
6. Esfarjani, K., Chen, G. & Stokes, H. Heat transport in silicon from first-principles calculations. *Phys. Rev. B* **84**, 085204 (2011).
7. Lundstrom, M. *Fundamentals of Carrier Transport*. (Cambridge University Press, 2009).

REVIEWERS' COMMENTS

Reviewer #1 (Remarks to the Author):

Author have responded satisfactorily. The paper can be accepted.

Reviewer #2 (Remarks to the Author):

The authors have carefully revised the manuscript and now it is suitable for publication.

Reviewer #3 (Remarks to the Author):

I think the authors replied to all questions and comments raised by the referees reasonably well. Therefore, I happily recommend the publication of this work to Nature Communications.

Revision report for MS# NCOMMS-20-31595A

Direct observation of large electron-phonon interaction effect on phonon heat transport

Jiawei Zhou, Hyun D. Shin, Ke Chen, Bai Song, Ryan A. Duncan, Qian Xu, Alexei A. Maznev, Keith A. Nelson, Gang Chen

We thank all the reviewers for their positive comments. In this revised version we have mainly changed the manuscript and supplementary information to meet the format requirements. Changes are highlighted (in yellow). The revised manuscript and supplementary information are attached in this document, following our revision report.

Reviewer #1:

Author have responded satisfactorily. The paper can be accepted.

Reviewer #2:

The authors have carefully revised the manuscript and now it is suitable for publication.

Reviewer #3:

I think the authors replied to all questions and comments raised by the referees reasonably well. Therefore, I happily recommend the publication of this work to Nature Communications.

Response to all: We thank all reviewers for their positive and constructive comments.